# Optimization of Bioactive Compound Extraction from Eggplant Peel by Response Surface Methodology: Ultrasound-Assisted Solvent Qualitative and Quantitative Effect

**DOI:** 10.3390/foods11203263

**Published:** 2022-10-19

**Authors:** Shiva Shahabi Mohammadabadi, Mohammad Goli, Sara Naji Tabasi

**Affiliations:** 1Department of Food Science and Technology, Isfahan (Khorasgan) Branch, Islamic Azad University, Isfahan 81551-39998, Iran; 2Laser and Biophotonics in Biotechnologies Research Center, Isfahan (Khorasgan) Branch, Islamic Azad University, Isfahan 81551-39998, Iran; 3Department of Food Nanotechnology, Research Institute of Food Science and Technology (RIFST), Mashhad 139-91735, Iran

**Keywords:** eggplant peel waste, solvent optimal formula, total phenol content, delphinidin-3-glucoside, yield extract, antioxidant activity

## Abstract

Anthocyanin pigments, which the peel of eggplant is rich in, contribute to food quality because of their function in color, appearance, and nutritional advantages. For the first time, this study aimed to optimize the composition of the extracting solvent as three factors: factor A (ratio of ethanol to methanol 0–100% *v*/*v*), factor B (ratio of water to alcohol 0–100% *v*/*v*), and factor C (citric acid in the final solvent 0–1% *w*/*v*) using response surface methodology (RSM), central composite design (CCD) with α 2, and two repeats in axial and factorial points and four central points, for maximum total phenolic content, total anthocyanin content, extraction yield, antioxidant activity in terms of DPPH radical scavenging activity and ferric reducing antioxidant power (FRAP) assay of the eggplant peel dry extract assisted by ultrasound (200 watts power, frequency of 28 kHz) in 60 °C for 45 min has been investigated. The best optimal formulas determined using RSM for the final solvent comprised optimal formula 1 (i.e., ethanol-to-methanol ratio 59% and water-to-alcohol ratio 0%, and citric acid in final solvent 0.47%), and optimal formula 2 (i.e., ethanol-to-methanol ratio 67% and water-to-alcohol ratio 0%, and citric acid in final solvent 0.56%). In general, an alcoholic–acidic extract of eggplant peel made with an ethanol–methanol solvent including citric acid can be used in the food industry as a natural source of antioxidants and pigment.

## 1. Introduction

Eggplant fruits, *Solanum melongena* L., are extensively consumed across the world due to their high nutraceutical content. Many researchers have focused their efforts in recent years on the recovery of potent antioxidants from plant byproducts. The byproducts of vegetable and fruit processing are high in both primary and secondary metabolites, i.e., terpenes, alkaloids, and phenolic compounds. Phenolic compounds have received a lot of interest because of their qualities that can help with a number of diseases. Furthermore, polyphenols may have a role in lowering the risk of metabolic syndrome due to their strong antioxidant activity, which can activate other defensive mechanisms [1,2]. Anthocyanins are glycosides of polyhydroxy and polymethoxy derivatives of 2-phenyl benzopyrylium cation salts that offer a variety of health advantages, including antioxidant, anti-cancer, anti-cardiovascular disorders, anti-diabetes, and anti-inflammation [3]. Eggplant peels contain a variety of chemicals, the most significant of which are anthocyanins. The glycosides of delphinidin give eggplant peels their purple hue [1]. The purple eggplant fruits have a high concentration of anthocyanins (12.08 mg/100 g DW), with delphinidin-3-O-rutinoside accounting for nearly 90% of the total [3]. Delphinidin is the most abundant anthocyanin in eggplant peels, with delphinidin-3-(p-coumaroyl rutinoside)-5-glucoside (nasunin) and delphinidin-3-rutinoside being the most abundant delphinidin-glycosides [1].

The outer layers of eggplants contain anthocyanins. Their extraction is challenging due to their ability to attach to carbohydrates and proteins, as well as the fact that each plant matrix has a varied composition. Traditional extraction techniques can be used to recover phenolic compounds from various source materials or agro-industrial by-products [4]. Because of the prolonged heat treatment, these procedures have a low efficiency and it might result in the degradation of sensitive molecules as well as an increase in energy, time, and solvent usage. Anthocyanins are unstable and readily destroyed, particularly when heated. Polymeric chemicals may be formed while anthocyanin degradation proceeds. Several endogenous and exogenous variables influence anthocyanin degradation, including phenolics, protein, carbohydrate, light, temperature, oxygen, pH, metal ions, and so on [3].

Heat treatment is a common commercial method for extending and preserving the shelf life of products; however, it frequently results in considerable phenolic component losses. Many plant materials, including eggplant peels, have been extracted using solid–liquid extraction and ultrasound-assisted extraction [1]. Anthocyanins in aqueous solution are generally hydrolyzed to form aglycone, then converted to chalcone, and eventually degraded into 2,4,6-trihydroxybenzaldehyde and p-hydroxybenzoic acid or its derivatives. When metal ions are present in the solution, anthocyanins are oxidized to quinones, which subsequently react with amino acids, glucose, polyphenols, or other molecules to form condensation products [5].

The recovery of phenolic compounds from natural products or agro-industrial by-products can be accomplished by a variety of approaches, including conventional methods (Soxhlet, heat reflux, maceration). The fundamental disadvantage of these approaches is their low extraction efficiency. This is due to prolonged heat treatment, which causes thermo-sensitive chemicals to degrade/loss while also using a significant amount of energy, time, and solvent. The grinding procedure has also been utilized for phenolic compound extraction. This procedure, however, is inefficient, since the extracts include a high quantity of contaminants. As a result, purifying methods are required to eliminate the undesirable components [6]. In order to protect the stability of phenolic compounds, innovative extraction technologies such as ultrasounds, microwaves, pulsed electric fields, high-voltage electrical discharges, supercritical fluid extraction, and pressurized liquid extraction have recently been developed. As a consequence, these techniques demonstrate excellent extraction efficiency, superior final product quality at lower temperatures, quicker kinetics, and lower energy/solvent consumption. Ultrasounds and other so-called assisted procedures have lately been employed widely to maintain sensitive chemicals. Good extraction efficiency, high quality of the end product at a lower temperature, and reduced energy/solvent usage are the key benefits given by these approaches [7]. The sudden production of bubbles in a liquid, which grows and then collapses violently, sending high-speed jets of solvent toward the solid surface, is the acoustic cavitation mechanism. These jets will boost mass transfer and hence accelerate the diffusion process by increasing the contact surface area between the solid matrix and the solvent phase [8].

For the first time, this study aimed to optimize the composition of the extracting solvent as three factors: factor A (ratio of ethanol to methanol 0–100% *v*/*v*), factor B (ratio of water to alcohol 0–100% *v*/*v*), and factor C (citric acid in the final solvent 0–1% *w*/*v*) using response surface methodology, central composite design (CCD) with an alpha of 2, and two repeats in axial and factorial points and four central points, for maximum total phenolic content, total anthocyanin content, extraction yield, antioxidant activity in terms of DPPH radical scavenging activity and ferric reducing antioxidant power (FRAP) assay of the eggplant peel dry extract assisted by ultrasound (200 watts power, frequency of 28 kHz) in 60 °C for 45 min has been investigated. 

## 2. Materials and Method

### 2.1. Preparation of Samples for Extraction

The long-shaped purple eggplant fruit was harvested after maturity near Khorasgan, Isfahan, Iran, and was of the Ghasri eggplant variety. The purple eggplant fruits were first cleaned, then the skins were manually peeled from the flesh with a paring knife and dried in a 40 °C oven for 12 h before being ground to powder with an electric grinder (LG-01, Ruian Baixin Pharmaceutical Machinery factory Co., Ltd., Wenzhou, Zhejiang, China).

### 2.2. Extraction Procedure and Yield

A total of 3 g of eggplant peel powder was mixed with 30 mL of combined solvent to make the sample preparation for the extraction process. The sample was treated in a 200-watt ultrasonic bath with a sonication frequency of 28 kHz, a temperature of 60 °C, and a 45 min extraction period. A power supply (HAMEG HM8150, Darmstadt, Hesse, Germany) was used in conjunction with an ultrasonic transducer to achieve the sonication. The extraction solution was put in a thermostat bath (Lauda E200, Lauda-Königshofen, Baden-Württemberg, Germany) that allowed time and temperature to be controlled. After passing the extract through filter paper, it was filtered. Finally, the extracts were transferred to a rotating vacuum evaporator to remove the solvent, after which the remaining solvent was evaporated using a water bath at 45–50 °C and dried. The total amount of dry matter extracted was calculated as a percentage of extraction yield based on the initial weight of the consumed eggplant peel [9]. Solid-phase extraction using Sep-Pak C18 columns cartridges (Sep-Pak Waters, Milford, MA, USA) according to Todaro et al. [10] with slight modification was used to purify the extract and remove interfering substances (i.e., reductive compounds, such as organic acids, sugars, and soluble proteins, etc.) before using subsequent reagents (i.e., Folin–Ciocalteu, 2,2-Diphenyl-1-picrylhydrazyl, and 2,4,6-Tripyridyl-s-triazine) in order to perform other tests, such as assessment of the total phenol and anthocyanin content, and antioxidant activity based on DPPH and FRAP. To keep light out, the lids were closed and covered with aluminum foil. The plates were placed in the freezer at −18 °C until the next tests were performed. 

### 2.3. Total Monomeric Anthocyanin Pigment Determination

The color of monomeric anthocyanin pigments changes reversibly with pH; the colorful oxonium form occurs at pH 1.0, while the colorless hemiketal form predominates at pH 4.5. The difference in pigment absorbance at 520 nm is related to pigment concentration. The results are given in terms of delphinidin-3-glucoside. Degraded anthocyanins in polymeric form are color-resistant regardless of pH, and are not included in the measurements because they absorb at pH 4.5 as well as pH 1.0. At 520 and 700 nm, the absorbance of a test fraction diluted with pH 1.0 buffer (potassium chloride, 0.025M) and pH 4.5 buffer (sodium acetate, 0.4M) was measured. The diluted test parts were compared to a distilled water-filled blank cell. Absorbance was measured within 20–50 min of preparation. The anthocyanin pigment concentration, expressed as delphinidin-3-glucoside equivalents, was calculated as follows [11]: *Monomeric anthocyanin* (mg/g dry extraction) = (*A* × *MW* × *DF* × 1000)/(*ε* × L)
where *A* = [(*A* 520 nm − *A* 700 nm) in pH 1.0] − [(*A* 520 nm − *A* 700 nm) in pH 4.5; *MW* (molecular weight)] = 465 g/M for delphinidin-3-glucoside (Dpd-3-glc); *DF* = dilution factor; L = path length in cm; *ε* = 29,000 molar extinction coefficient in L × mol^−1^ × cm^−1^, for Dpd-3-glc; and 1000 = factor for conversion from g to mg.

### 2.4. Total Phenolic Content (TPC)

The TPC of the crude extract was colorimetrically determined using the Folin–Ciocalteu method. The resulting crude extract was diluted with a dilution factor of 200. Then, in duplicate, 1.0 mL of the extract aliquot was transferred into a test tube with a 1 mL transfer pipette and carefully mixed with 5.0 mL of Folin–Ciocalteu reagent that had been previously diluted 1:10 with distilled water. After 3 min of shaking, 4.0 mL of sodium carbonate (7.5%, *w*/*v*) was added and well-mixed. After allowing the mixtures to stand in the dark for 30 min, the absorbance was measured in a single beam UV-Vis spectrophotometer (Ocean Optics, Dunedin, FL, USA) at 765 nm against a blank of methanol pure solvent. TPC values were reported in mg Gallic acid equivalent (GAE)/g of extract on a dry basis. Each solution was diluted with the solvent used for extraction in order to enter the Lambert–Beer linear zone [9,12].

### 2.5. Antioxidant Activity

#### 2.5.1. DPPH Radical Scavenging Activity

One mL of 0.135 mM DPPH (i.e., 2, 2-Diphenyl-1-picrylhydrazyl) prepared in methanol was combined with 1.0 mL of aqueous extract containing 0.2–0.8 mg/mL. The reaction mixture was completely vortexed and kept in the dark at room temperature for 30 min. At 517 nm, the absorbance was determined spectrophotometrically. The scavenging ability of the plant extract was calculated as the following [13]:DPPH scavenging activity (%) = [(Abs Control − Abs Sample)/(Abs Control)] × 100
where Abs Control indicates the absorbance of DPPH + methanol and Abs_sample_ indicates the absorbance of DPPH radical + sample (i.e., extract or standard).

#### 2.5.2. Ferric Reducing Antioxidant Power (FRAP) Assay

A sample (5 µL, 10–1000 g/mL) was allowed to react with 180 µL of ferric-TPTZ (i.e., 2,4,6-Tripyridyl-s-triazine) reagent [made by combining 300 mM acetate buffer, pH 3.6, 10 mM TPTZ in 40 mM HCl, and 20 mM FeCl_3_.6H_2_O in a 10:1:1 (*v*/*v*/*v*) ratio]. Additionally, the mixture was held at 37 °C for 5 min. A Thermo Varioskan Flash Microplate Reader was used to detect absorbance at 593 nm (Thermo Scientific, Waltham, MA, USA). Between 0.15 and 5.00 mM FeSO_4_, the standard curve was linear. The outcome was presented as FeSO_4_ levels established by standard curves. All samples were examined three times [14].

### 2.6. Experimental Design and Statistical Analysis

Response surface methodology (RSM) in statistics investigates the relationships between many causal variables and one or more response variables. The primary principle of RSM is to employ a series of well-prepared tests to arrive at an optimal response. This statistical design aids in reducing the number of experiments required to determine all quadratic regression model coefficients, as well as the components’ interactional effects [15]. Independent variables were included (A: ethanol-to-methanol ratio; from 0 to 100%), (B: water-to-alcohol ratio; from 0 to 100%) and (C: citric acid in final solvent; from 0 to 1%), which affected dependent variables such as total phenol content, anthocyanin content, and extraction yield, antioxidant activity based on DPPH radical scavenging activity and ferric reducing antioxidant power (FRAP) assay (Table 1). The quadratic and cubic polynomial model coefficients and optimization were calculated using the program Design-Expert 8.1.3 (State-Ease Inc., Minneapolis, Minnesota, USA). *p* values less than 0.05 were considered statistically significant. The data was collected in triplicate and the findings were averaged. The regression coefficients (β) were calculated by fitting the experimental data to the second-order and third-order polynomial models. The following is the generalized second-order and third-order polynomial model used in the response surface analysis:Y = β_0_ + β_1_A + β_2_B + β _3_C + β_11_A^2^ + β_22_B^2^ + β_33_C^2^ + β_12_AB + β_13_AC + β_23_BC + β_111_A^3^ + β_222_B^3^+ β_333_C^3^ + β_122_AB^2^ + β_133_AC^2^ + β_233_BC^2^ + β_112_A^2^B + β_113_A^2^C+ β_223_B^2^C+ β_123_ABC

β_0_ (i.e., constant coefficient), β_1_, β_2_, and β_3_ (i.e., linear coefficients), β_11_, β_22_, and β_33_ (i.e., quadratic coefficients), β_111_, β_222_, and β_333_ (i.e., cubic coefficients), and lastly, β_12_, β_13_, β_23_, β_122_, β_133_, β_233_, β_112_, β_113_, β_223_, and, β_123_ (i.e., interactive coefficients) were used to represent the coefficients of the polynomial model [16,17]. The A represents the ethanol-to-methanol ratio in percent, the B represents the water-to-alcohol ratio in percent, and the C represents the citric acid in the final solvent (%). As dependent variables or responses, mean values of antioxidant activity based on DPPH and FRAP, total phenol content, anthocyanin content, and extraction yield were examined (Y). On the basis of data acquired from studies, linear relationships and a quadratic and cubic polynomial were developed to produce experimental models for predicting responses. Following that, these models were statistically examined in order to choose the best one. A good model is one that has the highest R^2^ value from a statistical point of view [18,19,20].

## 3. Results and Discussion

### 3.1. Fitting the Response Surface Models for Total Phenol Content (TPC) 

The ANOVA was used to assess the first-order polynomial model’s relevance (Table 2). A high F-value and a small *p*-value for each term in the models would indicate a more significant effect on the response variable. Table 2 lists the TPC coefficient values along with their corresponding *p* values. R^2^, R^2^-adj, pred-R^2^, adeq precision, and coefficient of variation (CV) for the TPC were, respectively, 89, 87, 85, 23.95, and 6.4% (Table 2). As a result, the first-order polynomial model was shown to be more appropriate for the TPC than other models. The BC-interactive and linear coefficients of ethanol-to-methanol ratio (A), water-to-alcohol ratio (B), and citric acid in final solvent (C) were significant (*p* < 0.001) (Table 2). The AB-interactive, AC-interactive, quadratic, and cubic coefficients were all determined to be insignificant (*p* > 0.05) (Figure 1a,b). A lack-of-fit test (*p* > 0.05) was used to determine the model’s fitness, which demonstrated the model’s ability to accurately predict the variation. The maximum extraction of phenolic compounds was seen at low levels of water-to-alcohol ratio and low levels of citric acid, as shown in Figure 1c. The interaction of two factors revealed that lowering the water-to-alcohol ratio improved the extraction of phenolic compounds at all levels of citric acid. When the percentage of citric acid was lowered, the extraction of phenolic compounds increased whenever the water-to-alcohol ratio was low. The extraction of phenolic compounds was improved by increasing the ethanol-to-methanol ratio. The extraction of phenolic compounds was reduced by increasing the ratio of water to alcohol and the quantity of citric acid in the final solvent (Figure 1a–c and Table 2). The total phenol content figures are perfectly compatible with the data in Table 2. In earlier investigations, organic solvents containing acidification were shown to be more effective at extracting anthocyanins from eggplant peels [6]. There is no particular extraction method that can be regarded as standard [21]. The extraction efficiency of phenolic compounds is affected by the chemical composition of the compounds, the extraction process utilized, sample particle size, storage conditions and duration, as well as the presence of interfering chemicals [22]. Phenolic extract is composed of a complex combination of phenols that are selectively soluble in different solvents. In this regard, the polarity of the solvent is important in improving phenol solubility [23]. Despite water’s excellent extraction efficiency due to its considerably higher polarity, methanol and ethanol extract more phenolic chemicals than water does. The dielectric constants of water, methanol, and ethanol are 80.1, 32.7, and 24.5 D, respectively. In reality, the water solvent has dissolved a greater number of extractable solids, although not all of these chemicals are phenolic. Extracts with a range of contaminants (e.g., organic acids, sugars, and soluble proteins, etc.) can be implicated in identifying and quantifying phenols when water is used as the only solvent [24]. Total phenol levels, flavonoids, and the quantity of antioxidants extracted from mint leaves are all affected by solvent extraction. For phenol extraction, acetone and ethanol (75%) were also considered to be the most effective solvents. This study also discovered that, compared to alcohol, large quantities of water had no influence on the extraction of phenolic chemicals. In this investigation, increasing the water-to-alcohol ratio had no effect on the extraction of phenolic compounds, indicating that it is not a suitable solvent for these chemicals. Additionally, because ethanol is less polar than methanol, the extraction of phenolic components increased at greater ethanol-to-methanol ratios as compared to lower ethanol-to-methanol ratios. This indicates that mixing these two solvents enhances the rate of extraction of these compounds and that they extract phenolic compounds with high agreement because the combined ethanol and methanol have a lower polarity than water [25]. Citric acid in low concentrations increases phenolic compound extraction, while high concentrations inhibit phenolic compound extraction. At pHs below 4, phenol oxidation is substantially accelerated, with diphen hydroquinone and p-benzoquinone being the first compounds to be discovered. As a result, a low pH indicates that phenols are oxidized more quickly [26]. Solvent type and concentration had an impact on total monomeric anthocyanin extraction as well as total phenolic content. As a consequence, their affinity for different solvents in terms of material solubilization of eggplant peel might be related to the solvent’s dielectric constant [9], as the dielectric constants of water, methanol, and ethanol are 80.1, 32.7, and 24.5 D, respectively. Consider the following findings for the two optimal solvent combinations suggested for the optimal formula-1; ethanol-to-methanol ratio (59%) and water-to-alcohol ratio (0%), and citric acid in final solvent (0.47%), and the optimal formula-2; ethanol-to-methanol ratio (67%) and water-to-alcohol ratio (0%), and citric acid in final solvent (0.56%). Because the water is fully eliminated in these two combined solvents, the extracts prepared from these two solvent compounds contain the highest phenolic content. Additionally, they had the highest antioxidant activity due to the phenolic compounds. Table 1 also clearly shows these findings.

### 3.2. Fitting the Response Surface Models for Anthocyanin Content 

The ANOVA was used to evaluate the significance of the third-order (i.e., cubic, polynomial) models (Table 3). For each term in the models, a large F-value and a small *p*-value would imply a more significant effect on the respective response variable. Table 3 shows each coefficient value of anthocyanin content with respective *p*-values. The values of R^2^, R^2^-adj, pred-R^2^, adeq precision, and CV for the anthocyanin content were 96, 95, 92, 31.84, and 6.79, respectively. This revealed the cubic model was more adequate than other models for the anthocyanin content. The linear coefficients for water-to-alcohol ratio (B) and citric acid in final solvent (C) factors were significant (*p* < 0.0001). The BC-, A^2^B-, AB^2^-interactive and quadratic coefficients were significant (*p* < 0.05) whereas the cubic coefficient was found to be insignificant (*p* > 0.05). The fitness of the model was investigated through a lack-of-fit test (*p* > 0.05), which indicated the suitability of models to accurately predict the variation. The quantity of anthocyanin extracted was unaffected by the ethanol-to-methanol ratio, and the efficiency of anthocyanin extraction improved significantly when the water-to-alcohol ratio decreased, as shown in Figure 2a,c, which is in accordance with the data in Table 2. Figure 2b indicates that while the ratio of ethanol to methanol had no effect on anthocyanin extraction from eggplant skin, the quantity of citric acid in the final solvent did, and the amount of anthocyanin extracted increased significantly as the amount of citric acid was increased. Figure 2c illustrates the highest and lowest anthocyanin extraction rates in the water-to-alcohol ratios of 0% and 100%, respectively, as well as the quantity of citric acid in the final solvent at 1% and 0%, respectively. The findings in Table 3 and the anthocyanin content figures are completely consistent. This study reveals that increasing water does not have a good effect on anthocyanin extraction and is not a suitable solvent for anthocyanin extraction, although citric acid does enhance the rate of anthocyanin extraction to some extent. It was recommended to employ a mildly acidic solution in the extraction of acylated anthocyanins to avoid hydrolysis. More and more diversified anthocyanins were extracted when a mildly acidic media was utilized. To obtain the form of flavylium cation that is red and stable in an acidic solution, acid must be used. Furthermore, anthocyanins have the greatest stability at pH = 1.8. In acidic media, anthocyanins are more stable than in neutral or alkaline media. Lowering the pH is attributable to the existence of considerably greater proportions of anthocyanins in the form of flavylium cation, and therefore the anthocyanin color is more stable at this pH, but its amount reduced as pH increased, resulting in color loss. For extracting anthocyanins from eggplant peels, normally solvents with acidification are used. This is used to improve extraction and prevent degradation, providing maximum isolation. These findings may explain why eggplant peels had the most phenolic chemicals in an acidic medium [6,27]. Das et al. [28] examined the influence of pH on the anthocyanin concentration of black and purple rice bran extracts. The findings of this investigation revealed that increasing the pH up to 2.5 resulted in a small rise in anthocyanin concentration. Furthermore, raising the pH (above 2.5) significantly decreased the monomeric anthocyanin concentration of rice bran extract. The production of colorless structures after pH = 2.5 causes a reduction in anthocyanin concentration in purple and black rice bran extract. Anthocyanins are mostly present in the peel of eggplant, with nasunins being the most abundant anthocyanins discovered here, but phenolic acids are the primary constituents in eggplant pulp, where anthocyanin concentration is quite low. Acidic mediums are more stable than neutral or alkaline mediums for anthocyanins. Lowering the pH is attributable to the existence of considerably greater proportions of anthocyanins in the form of flavylium cation, and therefore the anthocyanin color is more stable at this pH, but its amount reduced as pH increased, resulting in color loss [6].

### 3.3. Fitting the Response Surface Models for Extraction Yield 

The significance of the third-order (i.e., cubic, polynomial) models was determined using ANOVA (Table 4). A high F-value and a small *p*-value for each term in the models would indicate a more substantial influence on the response variable. Table 4 lists the extraction yield coefficient values along with their corresponding *p*-values. For the extraction yield, R^2^, R^2^-adj, pred-R^2^, adeq accuracy, and CV were 97, 96, 94, 35.80, and 3.47, respectively. As a result, the cubic model was shown to be superior to other models in terms of extraction yield. Significant linear coefficients were found for the ethanol-to-methanol ratio (A), water-to-alcohol ratio (B), and citric acid in final solvent (C) variables (*p* < 0.05). The AC-, A^2^C-interactive-, and quadratic coefficients were all significant (*p* < 0.05); however, the cubic coefficient was not (*p* > 0.05). A lack-of-fit test (*p* > 0.05) was used to determine the model’s fitness, which demonstrated the model’s ability to reliably predict variation. Figure 3a shows that the extraction yield increased as the water-to-alcohol ratio increased at all levels of the ethanol-to-methanol ratio, with the 50% ethanol-to-methanol ratio providing the maximum extraction yield at all levels of the water-alcohol ratio. According to Figure 3b, the extraction yield increased as the percentage of citric acid decreased at all levels of the ethanol-to-methanol ratio. The highest extraction yield was reported at 1% citric acid and 50% ethanol-to-methanol ratio. Figure 3c reveals that increasing the percentage of citric acid improved the extraction yield at all levels of water-to-alcohol ratio, and increasing the water-to-alcohol ratio increased the extraction yield at all levels of citric acid consumption. With a citric acid concentration of 1% and a water-to-alcohol ratio of 100%, the maximum extraction yield was found. The extraction yield figures are perfectly compatible with the data in Table 4. Extraction yield is affected by a number of factors, including solvent polarity and pH, extraction temperature and time, and sample composition. Solvent and sample components are the most relevant factors at the same extraction temperature and time. Water has a favorable influence on extraction yield, according to this study, and increasing the water-to-alcohol ratio improved extraction yield. In addition, when ethanol and methanol are used together, the extraction yield is higher than when they are used separately (maximum efficiency is when both alcohols are used approximately equally). The extraction of water-soluble and organic-soluble compounds is made easier by combining the use of water and organic solvents. This might explain why hydro-ethanol-methanol extraction efficiencies are higher than those of ethanol, methanol, and water alone. A considerable increase in water in the extraction system improves the quantity of proanthocyanidin extraction efficiency but reduces the extraction selectivity. The increase in water allows additional components in the eggplant peel to be extracted, lowering the quality of the hydroalcoholic extract. Water with polarity 1 has the best extraction performance among pure solvents, followed by methanol and ethanol with polarities of 0.762 and 0.654, respectively. The most efficient extraction was achieved with aqueous solution [29]. As the acidity rises, so does the mass transfer coefficient. As a consequence, as the concentration of citric acid decreased, the extraction efficiency of hydroalcoholic extract increased. The addition of water to the organic solvent improves extraction yield. By combining ethanol and water in a sufficient ratio, the polarity of both solvents can enable the extraction of all molecules that were soluble in both. This might explain why the yields of binary solvent systems are greater than mono-solvent yields. In earlier investigations, organic solvents containing acidification were also shown to be more effective for the extraction of anthocyanin from eggplant peels [27]. In ultrasound-assisted extraction, the cavitation mechanism comprises two basic steps: (i) dissolving of the target compounds by generating superficial tissue damage, i.e., rinsing or rapid extraction, and (ii) diffusion of the desirable chemicals into the extraction medium, i.e., slow extraction [6,30]. The mechanical effect, which produces a disruption of plant cell walls when cavitation bubbles collapse at the surface of the solid matrix, promotes an increase in yield during ultra-sonication. As a result, mass transfer is improved, and the contact surface area between the solvent and the plant material grows. Furthermore, a significant increase in the very local temperature improves analyte solubility in the solvent and facilitates their diffusion from the sample matrix to the outside area [9].

### 3.4. Fitting the Response Surface Models for Antioxidant Activity Based on DPPH Radical Scavenging Activity

The significance of the first-order polynomial models was assessed using ANOVA (Table 5). A large F-value and a small *p*-value for each term in the models would indicate a more substantial influence on the response variable. Table 5 lists the antioxidant activity coefficients and their corresponding *p*-values. The antioxidant activity R^2^, R^2^-adj, pred-R^2^, adeq accuracy, and CV values were 85, 82, 77, 16.14, and 10.18, respectively. As a result, the first polynomial model was shown to be superior to the other models in terms of antioxidant activity. Significant linear coefficients were found for the ethanol-to-methanol ratio (A), water-to-alcohol ratio (B), and citric acid in final solvent (C) variables (*p* < 0.01). The AB- and BC-interactive coefficients were significant (*p* < 0.01); however, the quadratic and cubic coefficients were not (*p* > 0.05). A lack-of-fit test (*p* > 0.05) was used to assess the model’s fitness, indicating the model’s ability to reliably predict variation. Figure 4a demonstrates that the antioxidant activity of the extract has improved with increasing the ratio of ethanol to methanol at each level of the water-to-alcohol ratio. The antioxidant activity of the extract decreased by increasing the quantity of water in the final solvent at each ratio of ethanol to methanol. The maximum antioxidant activity was seen at the ethanol-to-methanol ratio of 0% and the water-to-alcohol ratio of 0%, while the lowest antioxidant activity was recorded at the ethanol-to-methanol ratio of 0% and the water-to-alcohol ratio of 100%. Figure 4b reveals that the quantity of antioxidant activity of the extract decreased with increasing the proportion of citric acid in the final solvent at each level of the ethanol-to-methanol ratio. The antioxidant activity of the extract increased with increasing ethanol to methanol in the final solvent for each level of citric acid. The highest and lowest antioxidant activity in the generated extract was seen at 0% and 1% citric acid levels, and 100% and 0% ethanol-to-methanol ratios, respectively. Figure 4c shows that at a water-to-alcohol ratio of 100%, increasing the percentage of citric acid in the final solvent increased the antioxidant activity of the extract, whereas at a water-to-alcohol ratio of 0%, increasing the percentage of citric acid in the final solvent decreased the antioxidant activity of the extract. The highest and lowest antioxidant activity was observed in the produced extract at the level of 0% citric acid and in the ratio of water to alcohol of 0% and 100%, respectively. The antioxidant activity figures are perfectly compatible with the data in Table 5. The compatibility of solvent constituents is directly related to the extraction of organic materials from plant material. The extracted components can be easily extracted if they are well-polarized at the same polarity as the solvent; otherwise, they are difficult to extract. The high ethanol-to-methanol ratio boosted the rate of free radical scavenging DPPH in this investigation, making it more appropriate for extracting compounds with antioxidant characteristics. In addition, increasing the water used for eggplant peel extraction reduces the rate of free radical scavenging DPPH. It may be explained by the fact that water extraction includes a lot of impurities, which causes the number of non-phenolic chemicals to rise, lowering the antioxidant property [29]. Because of the antioxidant potential of different compounds with varied polarity, the antioxidant capacity of the extract has a strong relationship with the solvent utilized. The solubility of antioxidant chemicals from plant materials in the extraction solvent is crucial for antioxidant extraction. The quantity of phenolic compounds in plant substances is related to their antioxidant activity. Phenolic acids are free radical receptors that are formed from the chemical structure of benzoic and cinnamic acids. The capacity to take free radicals is enhanced by the presence of phenolic rings and side chains in the molecular structure [31]. For all solvents tested, anthocyanin extracts had the strongest antiradical activity when compared to phenolic extracts [32], which is consistent with the findings of our study, implying that raising the water-to-alcohol ratio reduces the amount of anthocyanin extracted and, as a result, the antioxidant properties of the extract.

### 3.5. Fitting the Response Surface Models for Antioxidant Activity Based on FRAP Assay

ANOVA was used to determine the significance of the third-order polynomial models (Table 6). For each term in the models, a high F-value and a small *p*-value would suggest a stronger effect on the response variable. The antioxidant activity coefficients and their corresponding *p*-values are listed in Table 6. R^2^, R^2^-adj, pred-R^2^, adeq accuracy, and CV values for antioxidant activity were 88, 84, 79, 14.94, and 11.34, respectively. As a result, the cubic model was shown to be superior to the other models in terms of antioxidant activity. The ethanol-to-methanol ratio (A) and the water-to-alcohol ratio (B) variables both had significant linear coefficients (*p* < 0.001). The AB-AC- and BC-interactive terms, as well as the quadratic terms of A and cubic terms of B, had a significant influence on antioxidant activity (*p* < 0.05), while the quadratic terms of B and C, as well as the cubic terms of A and B, did not (*p* > 0.05). A lack-of-fit test (*p* > 0.05) was performed to evaluate the model’s fitness, which indicates the model’s ability to predict variance reliably. According to Figure 5a, increasing the ethanol-to-methanol ratio at a level of 0% water-to-alcohol ratio improved the antioxidant activity of the extract. By increasing the quantity of water in the final solvent for each ethanol-to-methanol ratio, the antioxidant activity of the extract was lowered. The maximum and minimum antioxidant activity was observed at a 100% ethanol-to-methanol ratio and a 0% and 100% water-to-alcohol ratio, respectively. Figure 5b demonstrates that the antioxidant activity of the extract increased with increasing ethanol-to-methanol ratio at all levels of citric acid in the final solvent. At a 100% ethanol-to-methanol ratio, increasing the quantity of citric acid in the final solvent resulted in a considerable increase in antioxidant activity, but this trend was not detected at a 0% ethanol-to-methanol ratio. Figure 5c shows that increasing the percentage of citric acid in the final solvent increased the antioxidant activity of the extract at a water-to-alcohol ratio of 100%, whereas increasing the percentage of citric acid in the final solvent had decreasing effect on the antioxidant activity of the extract at a water-to-alcohol ratio of 0%. The antioxidant activities of dried eggplant extract were significantly reduced and increased by increasing the water-to-alcohol ratio at levels of 0% and 1% citric acid in the final solvent, respectively. The highest and lowest antioxidant activity was detected in the obtained extract at 0% citric acid, and 0% and 100% water-to-alcohol ratios, respectively. Increasing the amount of water used improves the extraction of compounds that do not have antioxidant characteristics and are thus unsuitable for phenolic compound extraction. High ethanol concentrations in the solvent can cause physical parameters like density, dynamic viscosity, and dielectric constant of the solvent to change, resulting in improved solubility of antioxidant chemicals. Because ethanol has a greater boiling point than methanol, it may be extracted for longer periods of time. For a plant in Thailand, extraction with 90% ethanol and 90% acetone had the largest quantity of phenolic compounds, free radical scavenging, and iron reduction power, indicating that the results of this investigation are consistent [33]. In general, low water levels, low percentages of citric acid, and high ethanol-to-methanol ratios demonstrated the best rate of iron reduction power in this investigation. Even when the extracts’ radical scavenger activity was measured by DPPH, extraction with acidified alcoholic solvent (ethanol: water: HCl; 70:30:1, v/v/v) appeared to be more efficient than extraction with malic acid solution or tartaric acid solution. In fact, although having comparable levels of total phenols (as assessed by the Folin–Ciocalteu technique), all three extracts showed varied radical scavenging activity. The acidified ethanolic extract, in particular, had the greatest radical scavenging efficacy. This indicates that its chemical makeup differed not just quantitatively, but also qualitatively. Furthermore, it may be that the phenols in the ethanol extract were endowed with chemical features with better radical scavenging efficiency [10].

### 3.6. Optimization of the Solvent Formulation and Survey of Actual and Predicted Data

The numerical optimization technique was used to optimize the solvent formulations when the weight and importance value for five responses were considered equal (Table 7). This study aimed to maximize the total phenol content (83.47–170.366 mg Gallic acid/g dry extract), anthocyanin content (0.659–4.94 mg delphinidin-3-glucoside/g dry extract), extraction yield (11–25%), antioxidant activity as assessed by DPPH radical scavenging activity (38.11–37.06%), and antioxidant activity as evaluated by ferric reducing antioxidant power assay (1286.64–3844.64 µM Fe^+2^/g dry extract) (Table 7). The best optimal formulas for the final solvent included the optimal formula 1 (i.e., ethanol-to-methanol ratio 59% and water-to-alcohol ratio 0%, and citric acid in final solvent 0.47%), and the optimal formula 2 (i.e., ethanol-to-methanol ratio 67% and water-to-alcohol ratio 0%, and citric acid in final solvent 0.56%) as the predicted results whose desirability values were equal to 0.95 and 0.82, respectively (Table 7). Additionally, the difference between predicted numbers (statistical software) and actual (performed in the laboratory) data was minimal.

## 4. Conclusions

Eggplant peel was studied as a natural source of phenolic compounds. As a result, the solvent used had a substantial impact on extraction efficiency and the amount of phenolic compounds extracted. Independent variables were included (A: ethanol-to-methanol ratio; from 0 to 100%), (B: water-to-alcohol ratio; from 0 to 100%) and (C: citric acid in final solvent; from 0 to 1%), which affected dependent variables such as total phenol content, anthocyanin content, extraction yield, and antioxidant activity based on DPPH radical scavenging activity and ferric reducing antioxidant power (FRAP) assay. The best optimal formulas selected by response surface methodology for the final solvent included the optimal formula 1 (i.e., ethanol-to-methanol ratio 59% and water-to-alcohol ratio 0%, and citric acid in final solvent 0.47%), and the optimal formula 2 (i.e., ethanol-to-methanol ratio 67% and water-to-alcohol ratio 0%, and citric acid in final solvent 0.56%) for maximizing the total phenol content, the anthocyanin content, the extraction yield, and the antioxidant activity of the eggplant peel extract samples. In general, an eggplant peel alcoholic–acidic extract prepared using an ethanol–methanol solvent including citric acid can be utilized in the food industry as a natural source of antioxidants and pigment.

## Figures and Tables

**Figure 1 foods-11-03263-f001:**
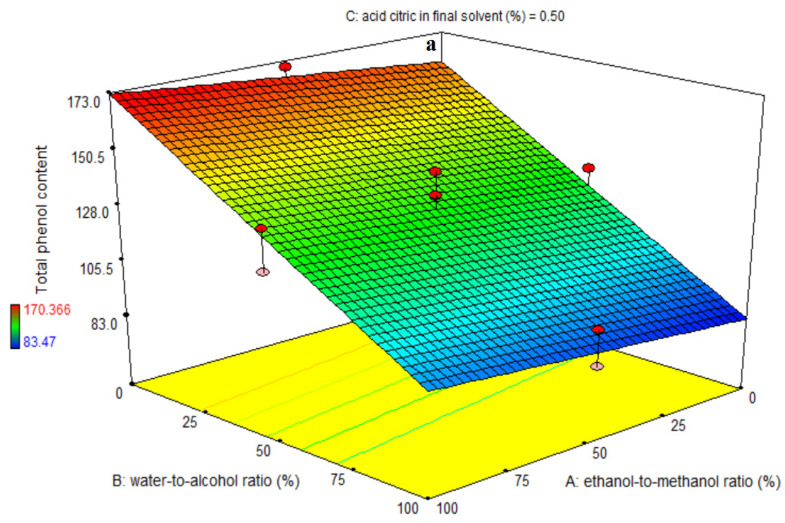
Effects of ethanol-to-methanol ratio (%), water-to-alcohol ratio (%) and citric acid in final solvent (%) on total phenol content (mg Gallic acid/g dry extract) in eggplant peel extraction. (**a**): The interaction effect of factors A and B on the intermediate level of factor C., (**b**): The interaction effect of factors A and C on the intermediate level of factor B., and (**c**): The interaction effect of factors B and C on the intermediate level of factor A.

**Figure 2 foods-11-03263-f002:**
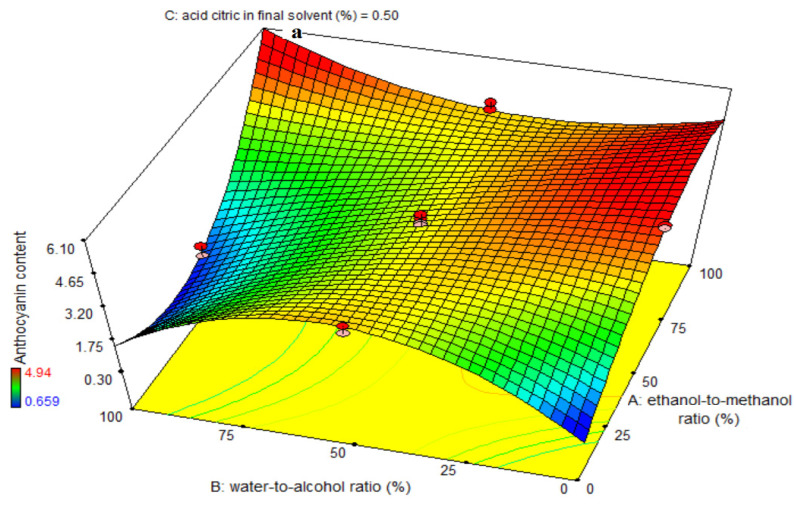
Effects of ethanol-to-methanol ratio (%), water-to-alcohol ratio (%) and citric acid in final solvent (%) on anthocyanin content (mg delphinidin-3-glucoside/g dry extract) in eggplant peel extraction. (**a**): The interaction effect of factors A and B on the intermediate level of factor C., (**b**): The interaction effect of factors A and C on the intermediate level of factor B., and (**c**): The interaction effect of factors B and C on the intermediate level of factor A.

**Figure 3 foods-11-03263-f003:**
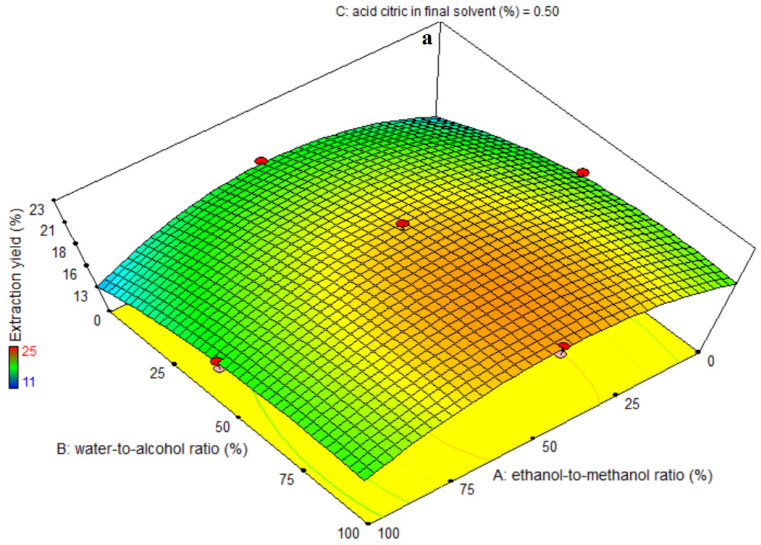
Effects of ethanol-to-methanol ratio (%), water-to-alcohol ratio (%) and citric acid in final solvent (%) on extraction yield (%) in eggplant peel extraction. (**a**): The interaction effect of factors A and B on the intermediate level of factor C., (**b**): The interaction effect of factors A and C on the intermediate level of factor B., and (**c**): The interaction effect of factors B and C on the intermediate level of factor A.

**Figure 4 foods-11-03263-f004:**
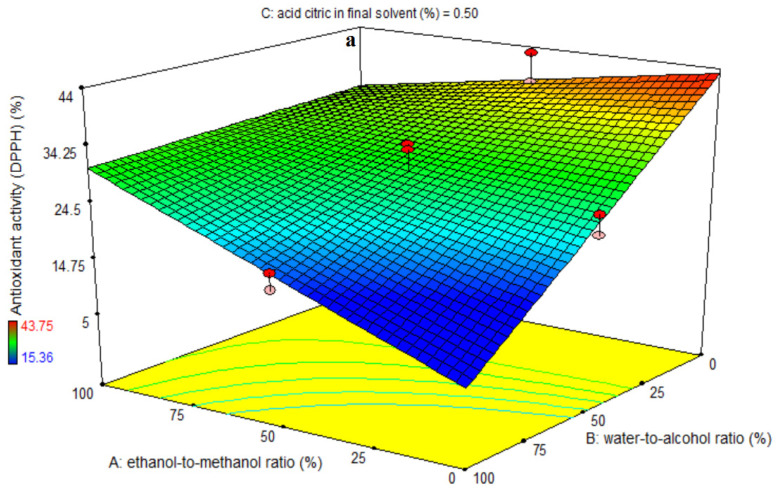
Effects of ethanol-to-methanol ratio (%), water-to-alcohol ratio (%) and citric acid in final solvent (%) on antioxidant activity in terms of DPPH radical scavenging activity (%) in eggplant peel extraction. (**a**): The interaction effect of factors A and B on the intermediate level of factor C., (**b**): The interaction effect of factors A and C on the intermediate level of factor B., and (**c**): The interaction effect of factors B and C on the intermediate level of factor A.

**Figure 5 foods-11-03263-f005:**
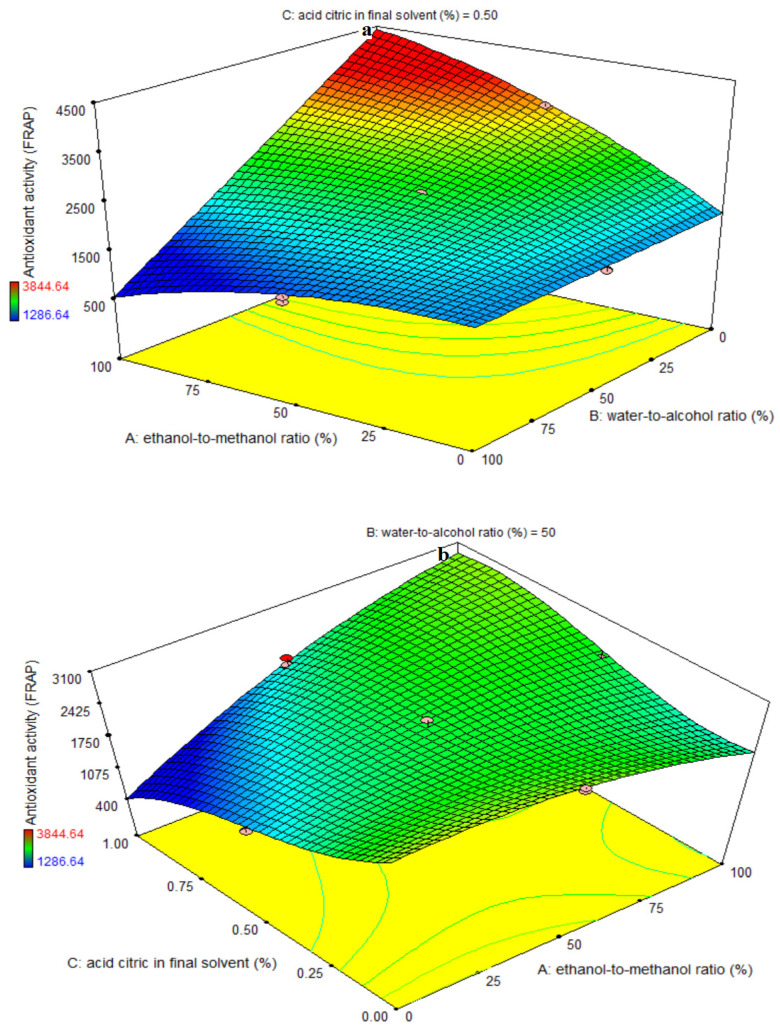
Effects of ethanol-to-methanol ratio (%), water-to-alcohol ratio (%) and citric acid in final solvent (%) on antioxidant activity in terms of ferric reducing antioxidant power (FRAP) assay (µM Fe^+2^/g dry extract) in eggplant peel extraction. (**a**): The interaction effect of factors A and B on the intermediate level of factor C., (**b**): The interaction effect of factors A and C on the intermediate level of factor B., and (**c**): The interaction effect of factors B and C on the intermediate level of factor A.

**Table 1 foods-11-03263-t001:** The uncoded and coded level of the independent variables and the composite central design (CCD) matrix, and the experimental data for the RSM responses in bioactive compound extraction of eggplant peel by the different solvent combinations.

Independent Variables				Code	Symbol				
					-α	−1	0	+1	+α
Ethanol-to-methanol ratio (%)				A	0	25	50	75	100
Water-to-alcohol ratio (%)				B	0	25	50	75	100
Citric acid in final solvent (%)				C	0	0.25	0.5	0.75	1
Run	Ethanol-to- Methanol Ratio (%)	Water-to-Alcohol Ratio (%)	Citric acid in Final Solvent (%)	Total Phenol Content *	Anthocyanin Content **	Extraction Yield (%)	Antioxidant Activity (%) *** (DPPH)		Antioxidant Activity (FRAP Assay) ****
1	0	50	0.5	115	3.79	19	22.26		1611
2	0	50	0.5	129	4.11	18	25.78		1631
3	25	75	0.75	92	3.09	23	17.96		1731
4	25	75	0.75	111	2.49	23	22.65		1925
5	25	75	0.25	93	1.96	20	17.96		2065
6	25	25	0.75	119	3.96	21	25.91		2105
7	25	25	0.75	125	4.1	20	25		2131
8	25	75	0.25	106	2.17	19	19.53		2345
9	25	25	0.25	155	3.75	18	33.98		2671
10	25	25	0.25	163	3.92	17	43.75		3365
11	50	100	0.5	98	1.13	23	18.22		1287
12	50	100	0.5	83	0.66	22	15.36		1378
13	50	50	1	119	4.26	25	23.17		1898
14	50	50	0.5	123	3.53	22	31.77		1991
15	50	50	1	116	3.67	24	27.34		2051
16	50	50	0.5	133	3.8	22	26.17		2125
17	50	50	0.5	133	3.97	21	25.52		2198
18	50	50	0.5	143	3.59	23	32.81		2438
19	50	50	0	151	2.61	12	32.68		2811
20	50	50	0	151	2.41	11	30.2		2878
21	50	0	0.5	164	4.94	18	37.63		3171
22	50	0	0.5	170	4.81	17	42.96		3371
23	75	75	0.25	100	2.49	19	26.43		1538
24	75	75	0.25	114	3.06	18	28.77		1818
25	75	75	0.75	117	3.62	23	29.16		2131
26	75	75	0.75	111	3.46	24	28.38		2411
27	75	25	0.75	145	4.36	20	30.2		3118
28	75	25	0.25	156	4.28	17	36.2		3285
29	75	25	0.75	137	4.54	21	27.6		3678
30	75	25	0.25	157	4.16	16	34.24		3845
31	100	50	0.5	120	4.37	18	29.29		2265
32	100	50	0.5	137	4.08	17	30.99		2318

* (mg Gallic acid/g dry extract), ** (mg delphinidin-3-glucoside/g dry extract), *** DPPH radical scavenging activity (%), **** Ferric reducing antioxidant power (FRAP) assay (µM Fe^+2^/g dry extract).

**Table 2 foods-11-03263-t002:** The analysis of variance (ANOVA) of total phenol content (first-order fitted polynomial model) in bioactive compound extraction of eggplant peel by the different solvent combinations.

Total Phenol Content (mg Gallic Acid/g Dry Extract)							
Source	Coefficient of Final Equation in Terms of Coded Factors	Sum of Squares	df	Mean Square	F Value	Prob > F	
Model	127.76	14,768.55	4	3692.14	55.31	<0.0001	significant
A-ethanol-to-methanol ratio (%)	2.99	285.95	1	285.95	4.28	0.0482	
B-water-to-alcohol ratio (%)	−19.34	11,968.29	1	11,968.29	179.29	<0.0001	
C-citric acid in final solvent (%)	−7.01	1572.4	1	1572.4	23.55	<0.0001	
BC	7.67	941.91	1	941.91	14.11	0.0008	
Residual	-	1802.4	27	66.76	-	-	
Lack-of-Fit	-	797.96	10	79.8	1.35	0.2816	not significant
Pure Error	-	1004.44	17	59.08	-	-	
Cor Total	-	16,570.95	31		-	-	
R^2^: 0.89, Adj-R^2^: 0.87, Pred-R^2^: 0.85, Adeq precision: 23.95, C.V.: 6.4%							
Total phenol content=127.76+2.99A−19.34B−7.01C+7.67BC							

**Table 3 foods-11-03263-t003:** The analysis of variance (ANOVA) of anthocyanin content (third-order fitted polynomial model) in bioactive compound extraction of eggplant peel by the different solvent combinations.

Anthocyanin Content (mg Delphinidin-3-Glucoside/g Dry Extract)							
Source	Coefficient of Final Equation in Terms of Coded Factors	Sum of Squares	df	Mean Square	F Value	Prob > F	
Model	3.71	30.87	8	3.86	69.53	<0.0001	significant
B-water-to-alcohol ratio (%)	−0.99	15.83	1	15.83	285.17	<0.0001	
C-citric acid in final solvent (%)	0.3	2.92	1	2.92	52.68	<0.0001	
BC	0.13	0.29	1	0.29	5.18	0.0325	
A^2^	0.09	0.26	1	0.26	4.71	0.0406	
B^2^	−0.21	1.41	1	1.41	25.32	<0.0001	
C^2^	−0.12	0.47	1	0.47	8.47	0.0079	
A^2^B	0.32	0.84	1	0.84	15.08	0.0008	
AB^2^	0.28	1.28	1	1.28	23	<0.0001	
Residual	-	1.28	23	0.056	-	-	
Lack-of-Fit	-	0.31	6	0.052	0.91	0.5083	not significant
Pure Error	-	0.97	17	0.057	-	-	
Cor Total	-	32.15	31		-	-	
R^2^: 0.96, Adj-R^2^: 0.95, Pred-R^2^: 0.92, Adeq precision: 31.84, C.V.: 6.79%							
$\mathrm{Anthocyanin}\;\mathrm{content}=3.71-0.99B+0.3C+0.13\mathrm{BC}+0.09A^{2}-0.21B^{2}-0.12{\;C}^{2}+0.32A^{2}B+0.28{\;\mathrm{AB}}^{2}$							

**Table 4 foods-11-03263-t004:** The analysis of variance (ANOVA) of extraction yield (third-order fitted polynomial model) in bioactive compound extraction of eggplant peel by the different solvent combinations.

Extraction Yield (%)							
Model	22.22	323.69	8	40.46	86.32	<0.0001	significant
A-ethanol-to-methanol ratio (%)	−0.22	1.53	1	1.53	3.27	0.0838	
B-water-to-alcohol ratio (%)	1.22	47.53	1	47.53	101.4	<0.0001	
C-citric acid in final solvent (%)	3.25	169	1	169	360.53	<0.0001	
AC	0.31	1.56	1	1.56	3.33	0.0809	
A^2^	−1	32	1	32	68.27	<0.0001	
B^2^	−0.5	8	1	8	17.07	0.0004	
C^2^	−1	32	1	32	68.27	<0.0001	
A^2^C	−1.31	13.78	1	13.78	29.4	<0.0001	
Residual	-	10.78	23	0.47	-	-	
Lack-of-Fit	-	2.28	6	0.38	0.76	0.6105	not significant
Pure Error	-	8.5	17	0.5	-	-	
Cor Total	-	334.47	31	-	-	-	
R^2^: 0.97, Adj-R^2^: 0.96, Pred-R^2^: 0.94, Adeq precision: 35.80, C.V.: 3.47%							
$\mathrm{Extraction}\;\mathrm{yield}=22.22-0.22A+1.22B+3.25C+0.31\mathrm{AC}-A^{2}-0.5B^{2}-C^{2}-1.31A^{2}C$							

**Table 5 foods-11-03263-t005:** The analysis of variance (ANOVA) of antioxidant activity in terms of DPPH radical scavenging activity (second-order fitted polynomial model) in bioactive compound extraction of eggplant peel by the different solvent combinations.

Antioxidant Activity in Terms of DPPH Radical Scavenging Activity (%)							
Source	Coefficient of Final Equation in Terms of Coded Factors	Sum of Squares	df	Mean Square	F Value	Prob > F	
Model	28.12	1219.03	5	243.81	29.74	<0.0001	significant
A-ethanol-to-methanol ratio (%)	1.83	107.75	1	107.75	13.14	0.0012	
B-water-to-alcohol ratio (%)	−5	800.6	1	800.6	97.64	<0.0001	
C-citric acid in final solvent (%)	−1.84	107.82	1	107.82	13.15	0.0012	
AB	2.19	76.74	1	76.74	9.36	0.0051	
BC	2.81	126.11	1	126.11	15.38	0.0006	
Residual	-	213.18	26	8.2	-	-	
Lack-of-Fit	-	64.47	9	7.16	0.82	0.6071	not significant
Pure Error	-	148.71	17	8.75	-	-	
Cor Total	-	1432.21	31		-	-	
R^2^: 0.85, Adj-R^2^: 0.82, Pred-R^2^: 0.77, Adeq precision: 16.14, C.V.: 10.18%							
Antioxidant activity (based on DPPH radical scavenging)=28.12+1.83A−5B−1.84C+2.19AB+2.81BC							

**Table 6 foods-11-03263-t006:** The analysis of variance (ANOVA) of antioxidant activity in terms of ferric reducing antioxidant power (FRAP) assay (third-order fitted polynomial model) in bioactive compounds extraction of eggplant peel by the different solvent combination.

Antioxidant Activity in Terms of FRAP Assay (µM Fe^+2^/g Dry Extract)							
Source	Coefficient of Final Equation in Terms of Coded Factors	Sum of Squares	df	Mean Square	F Value	Prob > F	
Model	2447.88	1.25 × 10^7^	7	1.78 × 10^6^	24.91	<0.0001	significant
A-ethanol-to-methanol ratio (%)	192.71	1.19 × 10^6^	1	1.19 × 10^6^	16.61	0.0004	
B-water-to-alcohol ratio (%)	−499.67	7.99 × 10^6^	1	7.99 × 10^6^	111.65	<0.0001	
AB	−238.75	9.12 × 10^5^	1	9.12 × 10^5^	12.74	0.0015	
AC	212.92	7.25 × 10^5^	1	7.25 × 10^5^	10.14	0.004	
BC	160.41	4.12 × 10^5^	1	4.12 × 10^5^	5.75	0.0246	
A^2^	−85.88	3.54 × 10^5^	1	3.54 × 10^5^	4.95	0.0358	
C^3^	−57.43	8.97 × 10^5^	1	8.97 × 10^5^	12.53	0.0017	
Residual	-	1.72 × 10^6^	24	71,559.64	-	-	
Lack of Fit	-	8.82 × 10^5^	7	1.26 × 10^5^	2.56	0.0535	not significant
Pure Error	-	8.36 × 10^5^	17	49,156.45	-	-	
Cor Total	-	1.42 × 10^7^	31	-	-	-	
R^2^: 0.88, Adj-R^2^: 0.84, Pred-R^2^: 0.79, Adeq precision: 14.94, C.V.: 11.34%							
$\mathrm{Antioxidant}\;\mathrm{activity}\;\left(\mathrm{based}\;\mathrm{on}\;\mathrm{FRAP}\;\mathrm{assay}\right)=2447.88+192.71A-499.67B-238.75\mathrm{AB}+212.92\mathrm{AC}+160.41\mathrm{BC}-85.88A^{2}-57.43C^{3}$							

**Table 7 foods-11-03263-t007:** The limitations utilized for optimization of bioactive compound extraction of eggplant peel by the different solvent combinations, and the optimal formula and data validation (actual and predicted) for total phenol content, anthocyanin content, extraction yield, and antioxidant activity in terms of DPPH radical scavenging activity and ferric reducing antioxidant power (FRAP) assay.

Name	Upper	Lower	Upper			
	Goal	Limit	Limit	Weight	Weight	Importance
A-ethanol-to-methanol ratio (%)	is in range	0	100	1	1	3
B-water-to-alcohol ratio (%)	is in range	0	100	1	1	3
C-citric acid in final solvent (%)	is in range	0	1	1	1	3
Total phenol content *	maximize	83.47	170.366	1	1	3
Anthocyanin content **	maximize	0.659	4.94	1	1	3
Extraction yield (%)	maximize	11	25	1	1	3
Antioxidant activity (DPPH) *** (%)	maximize	15.36	43.75	1	1	3
Antioxidant activity (FRAP) ****	maximize	1286.64	3844.64	1	1	3
Optimal formulas for final solvent						
1- Ethanol-to-methanol ratio (59%) and water-to-alcohol ratio (0%), and citric acid in final solvent (0.47%) with desirability 0.95						
Parameters	Predicted		Actual		Δ (%)	
Total phenol content *	170.37		165.05		3.12	
Anthocyanin content **	5.2		5.08		2.31	
Extraction yield (%)	17.14		16.63		2.98	
Antioxidant activity (DPPH) *** (%)	38.11		37.06		2.76	
Antioxidant activity (FRAP) ****	3716.92		3675.13		1.12	
2- Ethanol-to-methanol ratio (67%) and water-to-alcohol ratio (0%), and citric acid in final solvent (0.56%) with desirability 0.82						
Parameters	Predicted		Actual		Δ (%)	
Total phenol content *	163		161.07		1.18	
Anthocyanin content **	5.4		5.17		4.26	
Extraction yield (%)	17.83		16.99		4.71	
Antioxidant activity (DPPH) *** (%)	34.64		33.35		3.72	
Antioxidant activity (FRAP) ****	3802.79		3696.86		2.79	

* (mg Gallic acid/g dry extract), ** (mg delphinidin-3-glucoside/g dry extract), *** DPPH radical scavenging activity (%), **** ferric reducing antioxidant power (FRAP) assay (µM Fe^+2^/g dry extract).

## Data Availability

The data presented in this study are available on request from the corresponding author.

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
