# Peer review of "Optimization of Bioactive Compound Extraction from Eggplant Peel by Response Surface Methodology: Ultrasound-Assisted Solvent Qualitative and Quantitative Effect"

_foods, 2022, doi:10.3390/foods11203263_

Round 1
Reviewer 1 Report
"Optimization of bioactive compounds extraction from eggplant peel by response surface methodology: ultrasound-assisted solvent qualitative and quantitative effect" is an interesting study with practical implications. The following suggestions may help to improve the manuscript.
Line 104: "extracted extracts" please reformulate!
Line 142: "30 ml" should be replace with "30 mL". Same observation all over the manuscript: "ml" must be replaced with "mL" (lines: 180,182, 183 etc)
Line 198: For the DPPH method, authors should refer to the initial publication: Brand-Williams et al. 1995, instead of Afkhami et al., 2018; Aiyegoro et al., 2010.
Line 253: Concerning the anthocyanins content, that authors expressed as ** (g delphinidin-3-glucoside / Kg dry extract), I recommend to express as ** (mg delphinidin-3-glucoside / g dry extract), in order to have unitary, all variables reported to g of dry extract. The values remain the same.
Lines 276, 284, 347, 392, 396, 573: Please reformulate/correct
Line 301: I suggest to replace "75 percent" with "75%"
Author Response
Dear Prof.
We used reviewers and your insightful comments and we corrected parts of the article which were not meaningful (see following text) and the manuscript was entirely corrected according to the reviewer viewpoints. Please do not hesitate to ask us any questions about the submitted manuscript. This research, like many other scientific studies, has many weaknesses. I would like to express my deep gratitude to the hard-working referee of the Journal for his English language re-editing and wise-scientific re-judgment.
Sincerely yours,
Mohammad Goli
Hi dear editorial board and reviewer 1
Yours sincerely, thank you, the editor and reviewer 1, for your kind consideration for your scientific attention to my manuscript, for having read it, and for your valuable scientific and intelligent comments. I hope that by using your guidance, dear referees will attempt to refine the article and raise its scientific level. The yellow, green, and turquoise highlight in the uncleaned-revised manuscript related to the final reviewer 1, 2, and 3 proposed amendments, respectively.
Comments and Suggestions for Authors
"Optimization of bioactive compounds extraction from eggplant peel by response surface methodology: ultrasound-assisted solvent qualitative and quantitative effect" is an interesting study with practical implications. The following suggestions may help to improve the manuscript.
- Greetings and respect, dear referee: Thank you for your interest and very valuable comments. I hope that by implementing these comments, the quality level of the article will improve a lot and it will satisfy you as a valuable and scientific reviewer.
Line 104: "extracted extracts" please reformulate!
- Thank you so much for the wise comment and it was done according to your suggestion as the following text:
This procedure, however, is inefficient since the extracts include a high quantity of contaminants.
Line 142: "30 ml" should be replaced with "30 mL". Same observation all over the manuscript: "ml" must be replaced with "mL" (lines: 180,182, 183 etc)
- It was done according to your intelligent comment throughout the manuscript text.
Line 198: For the DPPH method, authors should refer to the initial publication: Brand-Williams et al. 1995, instead of Afkhami et al., 2018; Aiyegoro et al., 2010.
- Thank you so much. It was done i.e., omitted two references and added “Brand-Williams et al. 1995” citation. Please mention within the manuscript text.
Line 253: Concerning the anthocyanins content, that authors expressed as ** (g delphinidin-3-glucoside / Kg dry extract), I recommend to express as ** (mg delphinidin-3-glucoside / g dry extract), in order to have unitary, all variables reported to g of dry extract. The values remain the same.
- I am very pleased with your accuracy and intelligence in judging this study. All the units pointed to them have been modified as you correctly said.
Lines 276, 284, 347, 392, 396, 573: Please reformulate/correct
- Thank you so much for your comment and it was done according to the following:
- Line 276 was changed to “When the percentage of citric acid was lowered, the extraction of phenolic compounds increased whenever the water-to-alcohol ratio was low”.
- Line 284 was changed to “There isn't a particular extraction method that can be regarded as standard.”
- Line 347 was changed to “The findings in Table 3 and the anthocyanin content figures are completely consistent.”
- Line 392 was changed to “Fig 3a shows that the extraction yield increased as the water-to-alcohol ratio increased at all levels of the ethanol-to-methanol ratio, with the 50% ethanol-to-methanol ratio providing the maximum extraction yield at all levels of the water-alcohol ratio.”
- Line 396 was changed to “According to 3b, the extraction yield increased as the percentage of citric acid decreased at all levels of the ethanol to methanol ratio. The highest extraction yield was reported at 1% citric acid and 50% ethanol to methanol ratio.”
- Line 573 was changed to “This study aimed to maximize the total phenol content (83.47-170.366 mg Gallic acid / g dry extract), anthocyanin content (0.659-4.94 mg delphinidin-3-glucoside / g dry extract), extraction yield (11-25%), antioxidant activity as assessed by DPPH radical scavenging activity (38.11-37.06%), and antioxidant activity as evaluated by ferric reducing antioxidant power assay (1286.64-3844.64 µM Fe+2 / g dry extract) (Table 7).”
Line 301: I suggest to replace "75 percent" with "75%"
- Thank you so much for providing this suggestion. It was done throughout the manuscript. Please pay attention to the “revised foods-1842753” resubmitted.
In general, I am very grateful for your very wise and intelligent judgement. Certainly, every research is not free of problems and I hope to make more use of your valuable comments in future research.

Reviewer 2 Report
Abstract: Please include background.
Please edit tables.
Please improve the quality of figures of RSM. The characters cannot be read easily.
Please edit references according to journal guidelines.
Please include authors' contributions section.
Please correct typos in the text.
Author Response
Dear Prof.
We used reviewers and your insightful comments and we corrected parts of the article which were not meaningful (see following text) and the manuscript was entirely corrected according to the reviewer viewpoints. Please do not hesitate to ask us any questions about the submitted manuscript. This research, like many other scientific studies, has many weaknesses. I would like to express my deep gratitude to the hard-working referee of the Journal for his English language re-editing and wise-scientific re-judgment.
Sincerely yours,
Mohammad Goli
Hi dear editorial board and reviewer 2
Yours sincerely, thank you, the editor and reviewer 2, for your kind consideration for your scientific attention to my manuscript, for having read it, and for your valuable scientific and intelligent comments. I hope that by using your guidance, dear referees will attempt to refine the article and raise its scientific level. The yellow, green, and turquoise highlight in the uncleaned-revised manuscript related to the final reviewer 1, 2, and 3 proposed amendments, respectively.
Comments and Suggestions for Authors
- Greetings and respect, dear referee: Thank you for your interest and very valuable comments, I hope that by implementing these comments, the quality level of the article will improve a lot and it will satisfy you as a valuable and scientific reviewer.
Abstract: Please include background.
- Thank you so much for your high accuracy. It was done and a short background was added in the first of the abstract according to the following text:
Anthocyanin pigments, which are rich in the eggplant peel, contribute to food quality because of their function in color, appearance, and nutritional advantages.
Please edit tables.
- Thank you for your authoritative and intelligent judgment, dear reviewer. Since the previous tables 2 and 3 were very complicated and compact, these two tables were divided into 5 tables and the total number of tables in the article was increased to 7 in order to create a better understanding for the readers. In addition, in all the tables, scientific and writing errors were corrected according to the opinion of other dear reviewers. Please pay attention to the “revised foods-1842753” resubmitted.
Please improve the quality of figures of RSM. The characters cannot be read easily.
- Thank you for your wise judgement, dear reviewer. Since the previous figs 1 and 2 were very complicated and compact, these two figures were separated into 5 figures in order to improve the quality of viewing the characters for the readers. Please pay attention to the “revised foods-1842753” resubmitted.
Please edit references according to journal guidelines.
- Thank you so much dear reviewer. It was done according to journal guidelines either in the manuscript text or in the references list. Please pay attention to the “revised foods-1842753” resubmitted.
Please include authors' contributions section.
- It was done according to the following text. Please pay attention to the “revised foods-1842753” resubmitted.
Author Contributions
Conceptualization, Shiva Shahabi Mohammadabadi and Sara Naji Tabasi; Formal analysis, Shiva Shahabi Mohammadabadi and Sara Naji Tabasi; Investigation, Shiva Shahabi Mohammadabadi and Mohammad Goli; Methodology, Mohammad Goli; Project administration, Mohammad Goli; Software, Mohammad Goli and Sara Naji Tabasi; Supervision, Mohammad Goli.
Please correct typos in the text.
- To a large extent, the text of the article received many scientific and structural corrections due to the valuable comments of three dear referees. The typing of the text of the article was re-checked and the error was fixed in many cases. Thank you very much for your attention and time.
In general, I am very grateful for your very wise and intelligent judgement. Certainly, every research is not free of problems and I hope to make more use of your valuable comments in future research.

Reviewer 3 Report
This manuscript is about the optimization of phenolic extraction from eggplant by-product. The presented information should improve.
If apply for this journal, different practical applications should be provided
Manuscript should be provided according to the Instructions for authors of this prestigious Journal
It has been reported that the Folin-Ciocalteu assay is not a specific assay for determining phenolic compounds due to the presence of nonphenolic compounds, such as vitamin C, Cu1+, and other reductive compounds having an aromatic ring within the molecule (e.g., amino acids and peptides) that can also reduce the Folin-Ciocalteu reagent. This is confirmed in lines 291-298. Then, an explanation should be given to justify the use of this method and include this justification in the text, because this suggest that an overestimation in the quantification of these compounds was done.
Lines 291-298, Why this does not happen with other solvents or solvents conventions? Explain.
Line 184: use the correct symbol
Lines 307 -310: Why this effect?
Lines 317-319: How does this happen?
Lines 356-359: How do this deformation take place? Explain.
How do the authors eliminate the antioxidant effect of citric acid from the extracts?
A small introduction can be incorporate in the abstract section
It is evident that phenolics are not the unique compounds reacting in FRAP assay. Explain this phenomenon.
Author Response
Dear Prof.
We used reviewers and your insightful comments and we corrected parts of the article which were not meaningful (see following text) and the manuscript was entirely corrected according to the reviewer viewpoints. Please do not hesitate to ask us any questions about the submitted manuscript. This research, like many other scientific studies, has many weaknesses. I would like to express my deep gratitude to the hard-working referee of the Journal for his English language re-editing and wise-scientific re-judgment.
Sincerely yours,
Mohammad Goli
Hi dear editorial board and reviewer 3
Yours sincerely, thank you, the editor and reviewer 3, for your kind consideration for your scientific attention to my manuscript, for having read it, and for your valuable scientific and intelligent comments. I hope that by using your guidance, dear referees will attempt to refine the article and raise its scientific level. The yellow, green, and turquoise highlight in the uncleaned-revised manuscript related to the final reviewer 1, 2, and 3 proposed amendments, respectively.
Comments and Suggestions for Authors
This manuscript is about the optimization of phenolic extraction from eggplant by-product. The presented information should improve.
- Greetings and respect, dear referee: Thank you for your interest and very valuable comments, I hope that by implementing these comments, the quality level of the article will improve a lot and it will satisfy you as a valuable and scientific reviewer.
If apply for this journal, different practical applications should be provided
- Thank you so much. Yes you are right and the practical application was omitted from the manuscript text.
Manuscript should be provided according to the Instructions for authors of this prestigious Journal
- The article was prepared according to the instructions of the authors. Please pay attention to the “revised foods-1842753” resubmitted.
It has been reported that the Folin-Ciocalteu assay is not a specific assay for determining phenolic compounds due to the presence of nonphenolic compounds, such as vitamin C, Cu1+, and other reductive compounds having an aromatic ring within the molecule (e.g., amino acids and peptides) that can also reduce the Folin-Ciocalteu reagent. This is confirmed in lines 291-298. Then, an explanation should be given to justify the use of this method and include this justification in the text, because this suggest that an overestimation in the quantification of these compounds was done.
Lines 291-298, Why this does not happen with other solvents or solvents conventions? Explain.
- Thank you so much dear reviewer for providing the wise comment.
- The following text was added to “2.2. Extraction procedure and yield” title in materials and method section for the better conception and justify of overestimation prevention.
Solid-phase extraction using C18 ODS SPE cartridges (Sep-Pak Waters Milford, MA, USA) according to Todaro et al. [10] with slight modification was used to purify the extract and remove interfering substances (i.e., reductive compounds, such as organic acids, sugars, and soluble proteins) before using subsequent reagents (i.e., Folin-Ciocalteu, 2,2-Diphenyl-1-picrylhydrazyl, and 2,4,6-Tripyridyl-s-triazine) in order to perform other tests such as the assessment the total phenol and anthocyanin content, and antioxidant activity based on DPPH and FRAP.
- It was done according to your suggestion in discussion section as the follow.
In this regard, the polarity of the solvent is important in improving phenol solubility [23]. Despite water's excellent extraction efficiency due to its considerably higher polarity, methanol and ethanol extract more phenolic chemicals than water does. The dielectric constant of water, methanol, and ethanol are 80.1, 32.7, and 24.5 D, respectively. In reality, the water solvent has dissolved a greater number of extractable solids, although not all of these chemicals are phenolic. Extracts with a range of contaminants (e.g., organic acids, sugars, and soluble proteins, etc.) can be implicated in identifying and quantifying phenols when water is used as the only solvent [24].
Line 184: use the correct symbol
- After 3 minutes of shaking, 4.0 mL of sodium carbonate (5%, w/v) was added and well mixed.
Lines 307 -310: Why this effect?
- Thank you so much dear reviewer for the wise comment. It was amended according to the following text:
Additionally, because ethanol is less polar than methanol, extraction of phenolic components increased at greater ethanol-to-methanol ratios as compared to lower ethanol-to-methanol ratios. This indicates that mixing these two solvents enhances the rate of extraction of these compounds and that they extract phenolic compounds with high agreement because the combined ethanol and methanol have a lower polarity than water [25].
Lines 317-319: How does this happen?
- Thank you so much dear reviewer for intelligent comment. It was done as the following text:
Solvent type and concentration had an impact on total monomeric anthocyanin extraction as well as total phenolic content. As a consequence, their affinity for different solvents in terms of material solubilization of eggplant peel might be related to the solvent's dielectric constant [9]. As the dielectric constants of water, methanol, and ethanol are 80.1, 32.7, and 24.5 D, respectively. Consider the following findings for the two optimal solvent combinations suggested in Table 7: 1- Ethanol-to-methanol ratio (59%) & water-to-alcohol ratio (0%), and citric acid in final solvent (0.47%), and 2- Ethanol-to-methanol ratio (67%) & water-to-alcohol ratio (0%), and citric acid in final solvent (0.56%). Because the water is fully eliminated in these two combined solvents, the extracts prepared from these two solvent compounds contain the highest phenolic content. Additionally, they had the highest antioxidant activity due to the phenolic compounds. Table 1 also clearly shows these findings.
Lines 356-359: How do this deformation take place? Explain.
- Thank you so much for your scientific comment. It was amended as the following text:
In acidic media, anthocyanins are more stable than in neutral or alkaline media. Lowering the pH is attributable to the existence of considerably greater proportions of anthocyanins, in the form of flavylium cation, and therefore the anthocyanin color is more stable at this pH, but its amount reduced as pH increased, resulting in color loss. For extracting anthocyanins from eggplant peels, normally use solvents with acidification. This is done to improve extraction and prevent degradation, providing maximum isolation. These findings may explain why eggplant peels had the most phenolic chemicals in acidic medium [6, 27].
How do the authors eliminate the antioxidant effect of citric acid from the extracts?
- Please consider to the above text and once again in the following text:
Solid-phase extraction using C18 ODS SPE cartridges (Sep-Pak Waters Milford, MA, USA) according to Todaro et al. [10] with slight modification was used to purify the extract and remove interfering substances (i.e., reductive compounds, such as organic acids, sugars, and soluble proteins, etc.) before using subsequent reagents (i.e., Folin-Ciocalteu, 2,2-Diphenyl-1-picrylhydrazyl, and 2,4,6-Tripyridyl-s-triazine) in order to perform other tests such as the assessment the total phenol and anthocyanin content, and antioxidant activity based on DPPH and FRAP.
A small introduction can be incorporate in the abstract section
- Thank you so much for your high accuracy. It was done and a short background was added in the first of the abstract according to the following text:
Anthocyanin pigments, which are rich in the eggplant peel, contribute to food quality because of their function in color, appearance, and nutritional advantages.
It is evident that phenolics are not the unique compounds reacting in FRAP assay. Explain this phenomenon.
- Thank you so much for your consideration on our manuscript. Please mention to the following text which consider in our manuscript in “2.2. Extraction procedure and yield” for justifying and confirming your suggest.
Solid-phase extraction using C18 ODS SPE cartridges (Sep-Pak Waters Milford, MA, USA) according to Todaro et al. [10] with slight modification was used to purify the extract and remove interfering substances (i.e., reductive compounds, such as organic acids, sugars, and soluble proteins, etc.) before using subsequent reagents (i.e., Folin-Ciocalteu, 2,2-Diphenyl-1-picrylhydrazyl, and 2,4,6-Tripyridyl-s-triazine) in order to perform other tests such as the assessment the total phenol and anthocyanin content, and antioxidant activity based on DPPH and FRAP.
In general, I am very grateful for your very wise and intelligent judgement. Certainly, every research is not free of problems and I hope to make more use of your valuable comments in future research.

Round 2
Reviewer 3 Report
Second ROUND: Although some of my suggestions were heeded by the authors, some aspects are not technical and scientifically acceptable for presentation and discussion of the obtained results in this prestigious journal.
Manuscript is not presented (again) according to the Instructions for authors of this Journal.
Lines 117 – 123: The process described by Todaro et al. is about the identification of anthocyanins from eggplant peel using HPLC-MS. Then, Identification should be properly provided (by chromatogram and HPLC-MS assay data).
In addition, the use of the separated compounds must be clarified. How did the authors do the purified anthocyanins-rich extract recovery? (using the mentioned method and equipment)
Author Response
Dear Prof.
We used reviewers and your insightful comments and we corrected parts of the article which were not meaningful (see following text) and the manuscript was entirely corrected according to the reviewer viewpoints. Please do not hesitate to ask us any questions about the submitted manuscript. This research, like many other scientific studies, has many weaknesses. I would like to express my deep gratitude to the hard-working referee of the Journal for his English language re-editing and wise-scientific re-judgment.
Sincerely yours,
Mohammad Goli
Hi dear editorial board and reviewer 3
Yours sincerely, thank you, the editor and reviewer 3, for your kind consideration for your scientific attention to my manuscript, for having read it, and for your valuable scientific and intelligent comments. I hope that by using your guidance, dear referees will attempt to refine the article and raise its scientific level. The turquoise highlight in the uncleaned-revised manuscript related to the final reviewer 3 proposed amendments, respectively.
Comments and Suggestions for Authors
Second ROUND: Although some of my suggestions were heeded by the authors, some aspects are not technical and scientifically acceptable for presentation and discussion of the obtained results in this prestigious journal.
- Greetings and respect, dear referee: Thank you for your interest and very valuable comments, I hope that by implementing these comments, the quality level of the article will improve a lot and it will satisfy you as a valuable and scientific reviewer.
Manuscript is not presented (again) according to the Instructions for authors of this Journal.
- Thank you so much for your intelligent comment. Unfortunately, the file I provided to the journal and the file that was sent to you are different because I am unable to access any files from the site that are in the format and template of the journal, which was, of course, delivered to you. Because of this, I must submit you the modification as a simple Word file every time, which might attract your attention to the fact that it does not comply with the journal's author guidelines. I will however once more announce the modifications I made in order to at least conform to the journal's settings:
- Practical application was removed
- Abbreviation was removed
- Author contributions was added
- Typos was rechecked.
- References both in the text and in the list of references at the end of the article was adjusted according to the author's guidelines.
- Figures and tables were separated as much as possible according to your suggestion, dear judge
- I think some changes will be applied by the journal in the next steps after your approval, dear reviewer, and your satisfaction will be achieved
Lines 117 – 123: The process described by Todaro et al. is about the identification of anthocyanins from eggplant peel using HPLC-MS. Then, Identification should be properly provided (by chromatogram and HPLC-MS assay data). In addition, the use of the separated compounds must be clarified. How did the authors do the purified anthocyanins-rich extract recovery? (Using the mentioned method and equipment)
- I'm happy that the paper has a referee who can support it with such precision and intellectual knowledge. I need to make certain issues clear to you, referee, in response to what you said.
- 1- We did not aim to separate anthocyanins quantitatively and qualitatively, and as in the measurement formula, we only used the absorbent column used in the Todaro method to remove impurities and interfering compounds with laboratory reagents (including Folin-Ciocalteu, 2,2-Diphenyl-1-picrylhydrazyl, and 2,4,6-Tripyridyl-s-triazine). As in the measurement formula ????????? a??ℎ???????, you can see that with the molecular weight of 465 g/mol used as the basis for measuring and reporting the amount of anthocyanin, was Delphinidin-3-glucoside.
- 2- After passing the solvent containing phenolic compounds, especially anthocyanins, the solution coming out of the column, which is free of interfering substances, was used to perform subsequent tests including total phenol and anthocyanin content, and antioxidant activity based on DPPH and FRAP.
- 3- If these interfering substances were not taken, overestimation would definitely have occurred in the measurement of monomeric anthocyanin, total phenol, and antioxidant properties, but this problem (your worrying overestimation) is not seen in table’s number 1 and 7, which indicates the accuracy of the tests. Our future researchers can solve your concern by passing through absorbent columns, and maybe it can be a suitable reference for future research.
- 4- Unfortunately, the student who completed this thesis unfortunately lost his parents due to the corona virus and suffered a severe mental crisis. I request you to give this student a little relief and recovery by ignoring any minor problems that certainly any thesis can have scientific problems (refer to the next text).
- 5- Unfortunately, all the articles in the world have completely ignored the issue of overestimation caused by interfering compounds with laboratory reagents, and you were the only expert who paid attention to this delicate point, and in this sense, I am happy to meet you. I will be very happy if you provide your scientific solutions as a guide via email mgolifood@yahoo.com, so that I will definitely use your scientific guidelines in future research and I will definitely give you a scientific reference. I try to take advantage of your knowledge as a scientific advisor for my students' theses.
If you have a better suggestion for me to use in the text of the article, please let me know so that I can improve the quality of the article by modifying it.
Comments and Suggestions for Authors
This manuscript is about the optimization of phenolic extraction from eggplant by-product. The presented information should improve.
- Greetings and respect, dear referee: Thank you for your interest and very valuable comments, I hope that by implementing these comments, the quality level of the article will improve a lot and it will satisfy you as a valuable and scientific reviewer.
If apply for this journal, different practical applications should be provided
- Thank you so much. Yes you are right and the practical application was omitted from the manuscript text.
Manuscript should be provided according to the Instructions for authors of this prestigious Journal
- The article was prepared according to the instructions of the authors. Please pay attention to the “revised foods-1842753” resubmitted.
It has been reported that the Folin-Ciocalteu assay is not a specific assay for determining phenolic compounds due to the presence of nonphenolic compounds, such as vitamin C, Cu1+, and other reductive compounds having an aromatic ring within the molecule (e.g., amino acids and peptides) that can also reduce the Folin-Ciocalteu reagent. This is confirmed in lines 291-298. Then, an explanation should be given to justify the use of this method and include this justification in the text, because this suggest that an overestimation in the quantification of these compounds was done.
Lines 291-298, Why this does not happen with other solvents or solvents conventions? Explain.
- Thank you so much dear reviewer for providing the wise comment.
- The following text was added to “2.2. Extraction procedure and yield” title in materials and method section for the better conception and justify of overestimation prevention.
Solid-phase extraction using C18 ODS SPE cartridges (Sep-Pak Waters Milford, MA, USA) according to Todaro et al. [10] with slight modification was used to purify the extract and remove interfering substances (i.e., reductive compounds, such as organic acids, sugars, and soluble proteins) before using subsequent reagents (i.e., Folin-Ciocalteu, 2,2-Diphenyl-1-picrylhydrazyl, and 2,4,6-Tripyridyl-s-triazine) in order to perform other tests such as the assessment the total phenol and anthocyanin content, and antioxidant activity based on DPPH and FRAP.
- It was done according to your suggestion in discussion section as the follow.
In this regard, the polarity of the solvent is important in improving phenol solubility [23]. Despite water's excellent extraction efficiency due to its considerably higher polarity, methanol and ethanol extract more phenolic chemicals than water does. The dielectric constant of water, methanol, and ethanol are 80.1, 32.7, and 24.5 D, respectively. In reality, the water solvent has dissolved a greater number of extractable solids, although not all of these chemicals are phenolic. Extracts with a range of contaminants (e.g., organic acids, sugars, and soluble proteins, etc.) can be implicated in identifying and quantifying phenols when water is used as the only solvent [24].
Line 184: use the correct symbol
- After 3 minutes of shaking, 4.0 mL of sodium carbonate (5%, w/v) was added and well mixed.
Lines 307 -310: Why this effect?
- Thank you so much dear reviewer for the wise comment. It was amended according to the following text:
Additionally, because ethanol is less polar than methanol, extraction of phenolic components increased at greater ethanol-to-methanol ratios as compared to lower ethanol-to-methanol ratios. This indicates that mixing these two solvents enhances the rate of extraction of these compounds and that they extract phenolic compounds with high agreement because the combined ethanol and methanol have a lower polarity than water [25].
Lines 317-319: How does this happen?
- Thank you so much dear reviewer for intelligent comment. It was done as the following text:
Solvent type and concentration had an impact on total monomeric anthocyanin extraction as well as total phenolic content. As a consequence, their affinity for different solvents in terms of material solubilization of eggplant peel might be related to the solvent's dielectric constant [9]. As the dielectric constants of water, methanol, and ethanol are 80.1, 32.7, and 24.5 D, respectively. Consider the following findings for the two optimal solvent combinations suggested in Table 7: 1- Ethanol-to-methanol ratio (59%) & water-to-alcohol ratio (0%), and citric acid in final solvent (0.47%), and 2- Ethanol-to-methanol ratio (67%) & water-to-alcohol ratio (0%), and citric acid in final solvent (0.56%). Because the water is fully eliminated in these two combined solvents, the extracts prepared from these two solvent compounds contain the highest phenolic content. Additionally, they had the highest antioxidant activity due to the phenolic compounds. Table 1 also clearly shows these findings.
Lines 356-359: How do this deformation take place? Explain.
- Thank you so much for your scientific comment. It was amended as the following text:
In acidic media, anthocyanins are more stable than in neutral or alkaline media. Lowering the pH is attributable to the existence of considerably greater proportions of anthocyanins, in the form of flavylium cation, and therefore the anthocyanin color is more stable at this pH, but its amount reduced as pH increased, resulting in color loss. For extracting anthocyanins from eggplant peels, normally use solvents with acidification. This is done to improve extraction and prevent degradation, providing maximum isolation. These findings may explain why eggplant peels had the most phenolic chemicals in acidic medium [6, 27].
How do the authors eliminate the antioxidant effect of citric acid from the extracts?
- Please consider to the above text and once again in the following text:
Solid-phase extraction using C18 ODS SPE cartridges (Sep-Pak Waters Milford, MA, USA) according to Todaro et al. [10] with slight modification was used to purify the extract and remove interfering substances (i.e., reductive compounds, such as organic acids, sugars, and soluble proteins, etc.) before using subsequent reagents (i.e., Folin-Ciocalteu, 2,2-Diphenyl-1-picrylhydrazyl, and 2,4,6-Tripyridyl-s-triazine) in order to perform other tests such as the assessment the total phenol and anthocyanin content, and antioxidant activity based on DPPH and FRAP.
A small introduction can be incorporate in the abstract section
- Thank you so much for your high accuracy. It was done and a short background was added in the first of the abstract according to the following text:
Anthocyanin pigments, which are rich in the eggplant peel, contribute to food quality because of their function in color, appearance, and nutritional advantages.
It is evident that phenolics are not the unique compounds reacting in FRAP assay. Explain this phenomenon.
- Thank you so much for your consideration on our manuscript. Please mention to the following text which consider in our manuscript in “2.2. Extraction procedure and yield” for justifying and confirming your suggest.
Solid-phase extraction using C18 ODS SPE cartridges (Sep-Pak Waters Milford, MA, USA) according to Todaro et al. [10] with slight modification was used to purify the extract and remove interfering substances (i.e., reductive compounds, such as organic acids, sugars, and soluble proteins, etc.) before using subsequent reagents (i.e., Folin-Ciocalteu, 2,2-Diphenyl-1-picrylhydrazyl, and 2,4,6-Tripyridyl-s-triazine) in order to perform other tests such as the assessment the total phenol and anthocyanin content, and antioxidant activity based on DPPH and FRAP.
In general, I am very grateful for your very wise and intelligent judgement. Certainly, every research is not free of problems and I hope to make more use of your valuable comments in future research.
